# Object Grasp Control of a 3D Robot Arm by Combining EOG Gaze Estimation and Camera-Based Object Recognition

**DOI:** 10.3390/biomimetics8020208

**Published:** 2023-05-18

**Authors:** Muhammad Syaiful Amri bin Suhaimi, Kojiro Matsushita, Takahide Kitamura, Pringgo Widyo Laksono, Minoru Sasaki

**Affiliations:** 1Faculty of Information and Communication Technology, Universiti Tunku Abdul Rahman, Jalan Universiti, Bandar Barat, Kampar 31900, Malaysia; syaifulamri@utar.edu.my; 2Graduate School of Engineering, Gifu University, 1-1 Yanagido, Gifu 501-1193, Japankitamura.takahide.i5@s.gifu-u.ac.jp (T.K.); pringgo@ft.uns.ac.id (P.W.L.); 3Intelligent Production Technology Research & Development Center for Aerospace (IPTeCA), Tokai National Higher Education and Research System, Gifu 501-1193, Japan; 4Industrial Engineering, Faculty of Engineering, Universitas Sebelas Maret, Surakarta 57126, Indonesia

**Keywords:** EOG, gaze estimation, robot arm, object grasp, welfare robot

## Abstract

The purpose of this paper is to quickly and stably achieve grasping objects with a 3D robot arm controlled by electrooculography (EOG) signals. A EOG signal is a biological signal generated when the eyeballs move, leading to gaze estimation. In conventional research, gaze estimation has been used to control a 3D robot arm for welfare purposes. However, it is known that the EOG signal loses some of the eye movement information when it travels through the skin, resulting in errors in EOG gaze estimation. Thus, EOG gaze estimation is difficult to point out the object accurately, and the object may not be appropriately grasped. Therefore, developing a methodology to compensate, for the lost information and increase spatial accuracy is important. This paper aims to realize highly accurate object grasping with a robot arm by combining EMG gaze estimation and the object recognition of camera image processing. The system consists of a robot arm, top and side cameras, a display showing the camera images, and an EOG measurement analyzer. The user manipulates the robot arm through the camera images, which can be switched, and the EOG gaze estimation can specify the object. In the beginning, the user gazes at the screen’s center position and then moves their eyes to gaze at the object to be grasped. After that, the proposed system recognizes the object in the camera image via image processing and grasps it using the object centroid. The object selection is based on the object centroid closest to the estimated gaze position within a certain distance (threshold), thus enabling highly accurate object grasping. The observed size of the object on the screen can differ depending on the camera installation and the screen display state. Therefore, it is crucial to set the distance threshold from the object centroid for object selection. The first experiment is conducted to clarify the distance error of the EOG gaze estimation in the proposed system configuration. As a result, it is confirmed that the range of the distance error is 1.8–3.0 cm. The second experiment is conducted to evaluate the performance of the object grasping by setting two thresholds from the first experimental results: the medium distance error value of 2 cm and the maximum distance error value of 3 cm. As a result, it is found that the grasping speed of the 3 cm threshold is 27% faster than that of the 2 cm threshold due to more stable object selection.

## 1. Introduction

In recent years, assistive devices for people with motor difficulties have attracted attention. However, the operation panels of the conventional assistive devices are based on finger operation. So, people with tetraplegia, such as amyotrophic lateral sclerosis (ALS) patients, are paralyzed from neck to toe and are unable to use them themselves. This can make it harder for them to take care of themselves or their own survival, as constant assistance from other people becomes essential. Researchers such as Pinheiro et al. [1] and A. Bilyea et al. [2] have extensively discussed the disadvantages of people with tetraplegia and the possible solutions. As motor functions are limited, other alternatives such as bio-signals are considered as essential substitutes for control purposes. Therefore, researchers such as Mehrdad Fatourechi et al. [3] and Alexandre L.C. Bissoli et al. [4] are researching and developing an interface that uses the head’s eye movements as input instead of the movements of the fingers. A system with an end-effector controlled by eye movement is an ideal solution and target for future research.

One of them is a method that analyzes alternate current electro-oculography (AC EOG) signals, which are biological signals generated by eye movements, and estimates the direction of eye movements and the point of gaze [5,6]. So far, the estimation of the direction of eye movement is realized based on the positive and negative amplitudes of two channels of AC EOG signals [7,8]. The channels are used to stimulate the basic conceivable movements of the eye: vertical and horizontal. The amplitude difference between the channels has enabled the determination of 360-degree eye ocular direction. Applications such as computer interfaces [9,10,11,12,13], wheelchairs [14,15,16,17], and robots [18,19,20] using the same operation input as the cross-keys are demonstrated. As a notable example in robotic application, Eduardo Iáñez et al. [19] conducted an object displacement task using a robot arm where the eye’s direction controls the direction of the arm. This shows that we can guide a robot to perform a certain task using eye movement direction. However, this method requires a considerable series of eye directions as input parameters and another supportive mechanism (in this case, an RFID), to precisely assist in locating the object’s position and destination.

Concurrently, a different approach to eye direction for AC EOG has also been explored. It is a method that estimates eye movement by calculating the area of the amplitude [21,22,23]. This method is commonly known as EOG gaze estimation. Moreover, the method has also been constructed using the positive and negative states of the amplitude of the two channels of AC EOG signals. In robotic applications, Ilhamdi et al. [24,25] have applied EOG gaze estimation to 3D robot arm control by installing two cameras on the side and top of the target object. The processes include (1) the sagittal plane and position control of the arm tip via EOG gaze estimation based on the side camera image; (2) display switching via eye blink discrimination based on EOG; and (3) horizontal plane and position control via EOG gaze estimation based on the top camera image, allowing object pointing to be realized by the robot arm. However, this control is still inadequate for the application of object grasping. Since EOG signals are deformed when they pass through the skin and the EOG gaze estimation has intrinsic errors, it is difficult to achieve high spatial accuracy that enables stable object grasping. The position of the robot arm and object center need to be precise. 

Accordingly, a calibration technique for AC-EOG-based gaze estimation is required in order to grasp an object using a robot arm. Several unique calibration techniques have been researched and developed. K. Sakurai et al. [26] proposed estimation improvement by combining EOG and Kinect. Kinect is used for the motion detection of the eye’s pupil in order to locate the gazing position, and both data from EOG and Kinect provided better precision. Then, M. Yan et al. [27] proposed a fuzzy mathematical model to improve precision levels. The user is required to gaze at several target points on a screen. The data are then used to determine the conversion parameters from eye moment and gaze estimation at the calibration phase. Furthermore, it can be adjusted to individual differences. These techniques greatly improve gaze estimation; however, a simpler method can be implemented for object grasping. 

Therefore, this study aims to realize object grasping in 3D robot arm control using EOG gaze estimation. In other words, since EOG signals are essentially information-deficient, we introduce an object recognition function based on camera image analysis to develop a system that can supplement errors in EOG gaze estimation and verify the performance.

## 2. Proposed System

To investigate real-time object grasping using EOG, a 3D robot workspace is implemented. Figure 1 shows the conceptual system. It consists of a robot arm with a gripper composed of four Dynamixel AX-12 servomotors (Robotis Co., Ltd., Seoul, Republic of Korea), two USB cameras for taking pictures from above and from the side of the robot arm, a bio-signal measurement device (two channels) for EOG signals, a PC for analyzing EOG signals (Microsoft Visual Studio 2017 C++ software), a display for showing the images from the USB cameras, and two grasping objects (one is a red cube and the other is a blue cube). The software consists of EOG estimation, object recognition for camera images, and robot arm control based on inverse kinematics.

### 2.1. EOG Measurement Method

The bio-signal measurement system consists of five disposable electrodes, two bio-signal sensors, and a data acquisition device (National Instruments USB-6008), as shown in Figure 2. The electrodes are attached to the eye’s periphery, and the horizontal eye movement can be measured by Ch1 signals, and the vertical eye movement can be measured by Ch2 signals. Figure 3 shows the configuration of the electrodes on the face. The bio-signal sensor (measurement circuit) is configured as a band-pass filter (low-cut frequency: 1.06 Hz, high-cut frequency: 4.97 Hz, and gain: 78 dB) to measure AC-EOG signals. The schematic of the sensor is shown in Figure 4. EOG can be divided into two types, DC-EOG and AC-EOG, depending on the band-pass setting values. The one used in this study is AC-EOG. Then, at a sampling rate of 2 kHz, the data acquisition device converts the voltage signal into digital data. The data are transmitted to a PC every two seconds to be handled as numerical values on a PC using Microsoft Visual Studio 2017 C++.

The EOG system is configured as two channels (Ch1 and Ch2). Based on Table 1, the system has a bandpass filter with a lower cut-off frequency of 1.06 Hz and an upper cut-off frequency of 4.97 Hz. This filter is used to convert the DC-EOG to AC-EOG. The gain for the system is 78 [db] with a sampling frequency of 1 kHz in the data acquisition device. Figure 5 and Figure 6 show the Bode plots of the Ch1 and Ch2 amplifier circuits, respectively. The upper figure is the gain diagram, which shows the change in gain magnitude with frequency, and the lower figure is the phase diagram, which shows the phase change with frequency. The amplification characteristics in the design are shown. These Bode plots characterize a bandpass filter with a lower cutoff frequency of 1.06 Hz and an upper cutoff frequency of 4.97 Hz. 

### 2.2. EOG Gaze Estimation Method

The AC-EOG gaze position can be estimated by analyzing the positive and negative amplitudes and the area (integral value) of the two AC-EOG amplitudes every two seconds. For the direction of eye movement, positive and negative thresholds are set, and then the positive and negative states are determined, by which the threshold the EOG signal exceeds. The amplitude of the EOG Ch1 signal corresponds to horizontal eye movements, and it can be used to determine left and right movements. The amplitude of the EOG Ch2 signal corresponds to vertical eye movements and can be used to determine up and down movements. The amount of eye movement is calculated by the integral value of the AC-EOG amplitude: Ch1 is the vertical eye movement and Ch2 is the horizontal eye movement.
(1)$${\mathrm{EOG}}_{{\mathrm{integral}}_{{\mathrm{ch}}_{i}}}=\left|\overset{}{\underset{th\_p}{\int }}{\mathrm{EOG}}_{{\mathrm{Ch}}_{i}}\left(t\right)dt\right|+\left|\overset{}{\underset{th\_n}{\int }}{\mathrm{EOG}}_{{\mathrm{Ch}}_{i}}\left(t\right)dt\right|$$
(2)$$\mathrm{th}\_\mathrm{positive}=\left\{t:{\mathrm{EOG}}_{{\mathrm{Ch}}_{i}}\left(t\right)>th+\right\}$$
(3)$$\mathrm{th}\_\mathrm{negative}=\left\{t:{\mathrm{EOG}}_{{\mathrm{Ch}}_{i}}\left(t\right)>th-\right\}$$
(4)i=1, 2

The visual point (x-coordinate from ch1 and y-coordinate from ch2) is estimated by combining these movement directions and amounts. In order to match the position of the displayed image with the estimated EOG gaze point, it is necessary to measure the two EOG signals when the user gazes at a specific point during calibration and establish an equal relationship between the EOG amplitude and the image size. The distance between the user’s eyes and the screen is fixed at 35.0 cm. Figure 7 and Figure 8 show the eye gaze configuration and the EOG gaze estimation.

### 2.3. Integration Algorithm between Camera Object Recognition and EOG Gaze Estimation

Previous research has stated that an error occurs between the estimated EOG gaze position [8,9] and the true object position. Since the error is caused by the degradation of EOG signals and is essentially impossible to recover, using object recognition for camera images should be an effective method to compensate for this error. As an object recognition algorithm, the simplest object recognition of the HSV method is applied to recognize two objects (a red cube and a blue cube). The essential algorithm for recognition is shown in Figure 9. Since the main purpose of this paper is to verify the combination function of EOG gaze estimation and object recognition, the HSV method extracts red and blue color areas from camera images, and the areas are binarized into black and white, and the centroid of each object is calculated, as shown in Figure 10. Then, the distance between the estimated EOG gaze position and the centroid of each object is calculated. Finally, the object which has the smallest distance within a certain distance (threshold) is judged as the target object to be grasped.

On the other hand, if the distance error exceeds the threshold, the system will redo the EOG gaze estimation. As a supplementary note, object recognition is suitable for determining grasp strategies since it is capable of not only grasping the position of the object but also the posture of the object, which leads to an effective grasping process for objects with complex shapes. Since the appropriate threshold value largely depends on the measurement environment, Experiment 1 clarifies the constructed system’s EOG gaze estimation error range. There are two types of threshold values: the medium error and the maximum error values. We verify them to discuss the appropriate threshold setting method.

### 2.4. Robot Arm Control Process

The robot arm with a gripper has a total of four degrees of freedom, as shown in Figure 11. Three degrees of freedom can control the robot limbs and one degree of freedom can control the gripper. The user can operate the 3D motion of the robot arm on two consecutive inputs of EOG gaze estimations for the top camera image (Figure 12: Up) and the side camera image (Figure 12: Down). The camera is automatically switched when the first EOG gaze estimation is completed. Then, when the object to be grasped is specified after the second EOG gaze estimation, inverse kinematics is performed using the object’s centroid as an input value, as shown in Figure 13. This value calculates the motor angles of the three degrees of freedom of the arm, and the robot arm motion is executed afterward.

### 2.5. Inverse Kinematics

Since the input of the actual machine is the joint angle, it is necessary to convert the target trajectory to the target angle of each joint. Since inverse kinematics is generally used for the trajectory generation of robot arms, inverse kinematics is derived below. 

#### Derivation of Inverse Kinematics

Figure 13 shows the coordinate system set for the flexible manipulator. All coordinate systems are left-handed, and the reference coordinate system is set at the base of the manipulator. We assume that the origins of $\Sigma_{0}$ through $\Sigma_{1}$ overlap. $l_{1},l_{2}$ are the distance between the axes of Joint2 and Joint3 and the distance from the axis of Joint3 to the tip, respectively. In addition, $\theta_{1},\theta_{2},\theta_{3}$ are the motion angles of Joint1, Joint2, and Joint3, respectively, and the upright state is 0[deg]. Normally, the posture of the robot arm should also be determined, but the manipulator used this time does not have enough degrees of freedom, so the tip posture is not considered.

If the coordinate transformation matrix from $\Sigma_{i-1}$ to $\Sigma_{i}$ is Tii−1, each transformation coordinate matrix is as follows.
(5)T10=cosθ1−sinθ100sinθ1cosθ10000100001
(6)T21=cosθ2−sinθ20000−10sinθ2cosθ2000001 
(7)T32=−sinθ3−cosθ300cosθ3−sinθ30l100100001
(8)TH3=100l2010000100001

Using the above, the transformation matrix from the reference coordinate system to the tip coordinate system is given in Equation (9).
(9)TH0=T10T21T32TH3=−cosθ1sinθ2+θ3−cosθ1cosθ2+θ3sinθ1−cosθ1l1sinθ2+l2sinθ2+θ3−sinθ1sinθ2+θ3−sinθ1cosθ2+θ3cosθ1−sinθ1l1sinθ2+l2sinθ2+θ3cosθ2+θ3−sinθ2+θ30l0+l1cosθ2+l2cosθ2+θ30001=Rd01

At this time, R represents the rotation coordinate and d represents the translation vector. In this study, we do not consider the rotating coordinate system. Only the translation vector d is used as a derivation of inverse kinematics. Again, the translation vector d is defined as follows.
(10)$$d=\left[\begin{array}{c} x\\ y\\ z \end{array}\right]=\left[\begin{array}{c} -\cos\theta_{1}\left\{l_{1}\sin\theta_{2}+l_{2}\sin\left(\theta_{2}+\theta_{3}\right)\right\}\\ -\sin\theta_{1}\left\{l_{1}\sin\theta_{2}+l_{2}\sin\left(\theta_{2}+\theta_{3}\right)\right\}\\ l_{1}\cos\theta_{2}+l_{2}\cos\left(\theta_{2}+\theta_{3}\right) \end{array}\right]$$

From the *x* and *y* components of Equation (10), the following equation holds.
(11)$$\frac{y}{x}=\tan\theta_{1}$$

Therefore, $\theta_{1}$ can be calculated as follows.
(12)$$\theta_{1}=\mathrm{atan}2\left(y,x\right)$$

Furthermore, the square of the vector length of Equation (10) is Equation (13).
(13)$$d^{2}=x^{2}+y^{2}+z^{2}=l_{1}^{2}+l_{2}^{2}+2l_{1}l_{2}\cos\theta_{3}$$

By transforming Equation (13), we obtain
(14)$$\cos\theta_{3}=\frac{x^{2}+y^{2}+z^{2}-l_{1}^{2}-l_{2}^{2}}{2l_{1}l_{2}}=D\;$$
(15)$$\sin\theta_{3}=\pm \sqrt{1-D^{2}}$$
(16)$$\tan\theta_{3}=\pm \frac{\sqrt{1-D^{2}}}{D}\;$$

From (16)
(17)$$\theta_{3}=\mathrm{atan}2\left(\pm \sqrt{1-D^{2}},D\right)$$

By transforming the *z*-direction of Equation (10),
(18)$$\begin{array}{ll} z & =\left(l_{1}+l_{2}\cos\theta_{3}\right)\cos\theta_{2}-l_{2}\sin\theta_{2}\sin\theta_{3}\\ & =\sqrt{{\left(l_{1}+l_{2}\cos\theta_{3}\right)}^{2}+{\left(l_{2}\sin\theta_{3}\right)}^{2}}\cos\left(\theta_{2}+\gamma \right)\\ & =\sqrt{l_{1}^{2}+l_{2}^{2}+2l_{1}l_{2}\cos\theta_{3}}\cos\left(\theta_{2}+\gamma \right)\\ & =\sqrt{l_{1}^{2}+l_{2}^{2}+2l_{1}l_{2}\frac{x^{2}+y^{2}+z^{2}-l_{1}^{2}-l_{2}^{2}}{2l_{1}l_{2}}}\cos\left(\theta_{2}+\gamma \right)\\ & =\sqrt{x^{2}+y^{2}+z^{2}}\cos\left(\theta_{2}+\gamma \right) \end{array}$$

However, the following holds for γ
(19)$$\begin{array}{c} \cos\gamma =\frac{l_{1}+l_{2}\cos\theta_{3}}{\sqrt{x^{2}+y^{2}+z^{2}}}\;\sin\gamma =\frac{l_{2}\sin\theta_{3}}{\sqrt{x^{2}+y^{2}+z^{2}}}\\ \gamma =\mathrm{atan}2\left(2l_{1}l_{2}\sin\theta_{3},x^{2}+y^{2}+z^{2}+l_{1}^{2}-l_{2}^{2}\right) \end{array}$$

From the above,
(20)$$\theta_{2}=\mathrm{atan}2\left(\pm \sqrt{x^{2}+y^{2}},z\right)-\mathrm{atan}2\left(2l_{1}l_{2}\sin\theta_{3}^{\prime },x^{2}+y^{2}+z^{2}+l_{1}^{2}-l_{2}^{2}\right)\;$$

From Equations (12), (17), and (20), each joint angle is determined by the following equation.
(21)$$\left[\begin{array}{c} \theta_{1}\\ \theta_{2}\\ \theta_{3} \end{array}\right]=\left[\begin{array}{c} \mathrm{atan}2\left(y,x\right)\\ \mathrm{atan}2\left(\pm \sqrt{x^{2}+y^{2}},z\right)-\mathrm{atan}2\left(2l_{1}l_{2}\sin\left(\theta_{3}+\beta \right),x^{2}+y^{2}+z^{2}+l_{1}^{2}-l_{2}^{2}\right)\\ \mathrm{atan}2\left(\pm \sqrt{1-D^{2}},D\right) \end{array}\right]$$

In Equation (21), there are two solutions because the manipulator can take two postures for any position (Figure 14). Therefore, this time, Equation (22), which is closer to the initial posture, is used as the solution, where $l_{0}=0.11\;\left[m\right],l_{1}=0.092\;\left[m\right],\;l_{2}=0.14\;m$
(22)$$\left[\begin{array}{c} \theta_{1}\\ \theta_{2}\\ \theta_{3} \end{array}\right]=\left[\begin{array}{c} \mathrm{atan}2\left(y,x\right)\\ \mathrm{atan}2\left(-\sqrt{x^{2}+y^{2}},z\right)-\mathrm{atan}2\left(2l_{1}l_{2}\sin\left(\theta_{3}+\beta \right),x^{2}+y^{2}+z^{2}+l_{1}^{2}-l_{2}^{2}\right)-\alpha \\ \mathrm{atan}2\left(-\sqrt{1-D^{2}},D\right)-\beta \end{array}\right]$$

## 3. Experiment 1: The Investigation of the Distance Error of EOG Gaze Estimation

The first experiment verifies the distance error of EOG gaze estimation (i.e., the gap between the estimated EOG gaze position and the target object’s centroid calculated from the camera object recognition). The procedure is as follows: first, the proposed camera object recognition algorithm calculates the centroids of red and blue cubes (height: 3.0 cm, length: 3.0 cm, and width: 3.0 cm), and then a hundred EOG gaze estimation trials are conducted for each of the two cubes, and the distance error is analyzed.

The results in Figure 15 indicate no significant difference in the distance error for the red and blue cubes, thus confirming that there is no effect of color and where the objects are positioned on the camera image. Overall, the distance error for the blue object has a mean value of 1.865 cm, a maximum value of 5.675 cm, and a minimum value of 0.171 cm, while the distance error for the red object has a mean value of 2.188 cm, a maximum value of 6.985 cm, and a minimum value of 0.1761 cm. It is confirmed that the distance error in our experimental setup ranges from 1.8 to 3.0 cm. Hence, Experiment 2 uses the intermediate distance error value of 2 cm and the maximum distance error value of 3 cm based on the above results. These values verify the appropriate threshold setting to perform with a higher speed and more stability in the proposed system.

## 4. Experiment 2: An Investigation of the Speed and Stability of the Proposed System

The second experiment evaluates the performance of the proposed system in terms of speed and stability for object grasping tasks at two different thresholds. The experiment is conducted on five subjects (age range: 20 to 40 years). The experiment conductor first explains the experiment’s significance and procedure, attaches EOG electrodes to the subject, and confirms that the subject can appropriately operate the system. 

In the experiment, the subject first needs to gaze at the top image’s center position and then gaze at the object to be grasped in the top image. Next, the image is switched to the side image when the proposed system appropriately estimates the first EOG gaze position of the subject. Later, the subject needs to see the side image’s center position and gaze at the object to be grasped in the side image, before the trial ends. Finally, the subject repeats eight times for each threshold setting; the first five trials are practice trials and the latter three trials are used for performance evaluation purposes.

As shown by the results in Table 2 and Figure 16, the average achievement time for threshold 2 cm is 71 s, while the average achievement time for threshold 3 cm is 52 s. This comparison means that the distance error is more appropriate for threshold 3 cm than for threshold 2 cm, resulting in a series of object-grasping operations with 27% faster speed. The narrow thresholds make the system more precise in selecting the object. However, the user needs undivided concentrations for eye gazing to perform the control smoothly.

## 5. Conclusions

This study aims to improve the object-grasping performance of 3D robot arm control based on EOG gaze estimation for people with tetraplegia. In conventional research, EOG gaze estimation is problematic in the sense that object grasping inconsistency fails due to errors and the inability to specify objects with high spatial accuracy. This paper introduces a new method of object recognition based on camera images to compensate for the EOG gaze estimation errors and achieve stable object grasping. The proposed system consists of two cameras installed on the top and side of the robot arm, which are viewed using a computer display. In the investigation experiment with five subjects, it is found that the error of EOG gaze estimation in the proposed system configuration ranges between 1.8 cm and 3.0 cm. Next, two threshold values based on EOG gaze estimation errors from the first experiment are set. Finally, an object-grasping experiment with a robot arm using a control input combining camera object recognition and EOG gaze estimation is conducted. As a result, when the threshold value is set to a medium error value of 2.0 cm, the robot arm fails to specify the object, taking 71 s on average to grasp the object. On the other hand, when the maximum error value of 3.0 cm is set as the threshold, the object selection is stable, and the average time is 52 s, i.e., the grasping time is 27% faster than the case where the medium error value is set as the threshold. In conclusion, it is proven that the introduction of camera object recognition can compensate for the error of the EOG gaze estimation and realize precise object grasping. In contrast, the conventional EOG gaze estimation has error values of 1.8 cm to 3.0 cm. There are advantages and disadvantages for the proposed system based on these results. The advantages of the system are as follows: (1) the system enables the remote controlling of a robot arm using the EOG method, (2) the implementation of the simple image processing method improves the EOG gaze estimation by targeting the center point of the object; and (3) the robot gripper is able to grab the target object successfully in order to conduct displacement tasks. As for the disadvantages, (1) the user is constrained to stay still and positioned 0.35 m from the computer display and (2) the condition of the captured image from camera can affect the accuracy of the image processing, e.g., the image brightness from room lighting and image pixel quality.

## Figures and Tables

**Figure 1 biomimetics-08-00208-f001:**
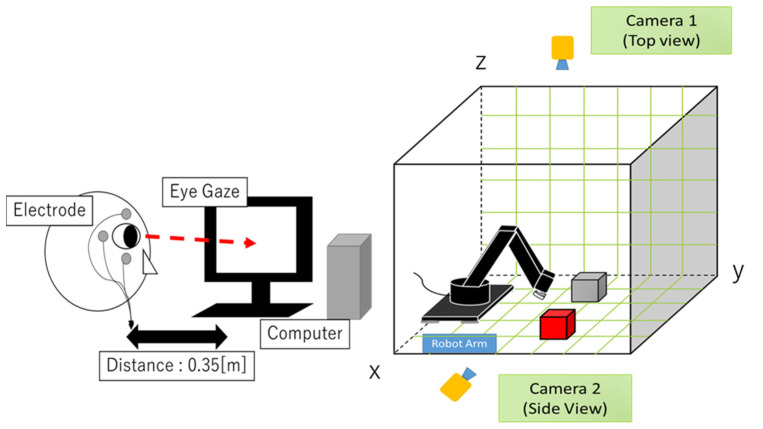
Proposed system: robot arm controlled by the combination of EOG gaze estimation and camera object recognition.

**Figure 2 biomimetics-08-00208-f002:**
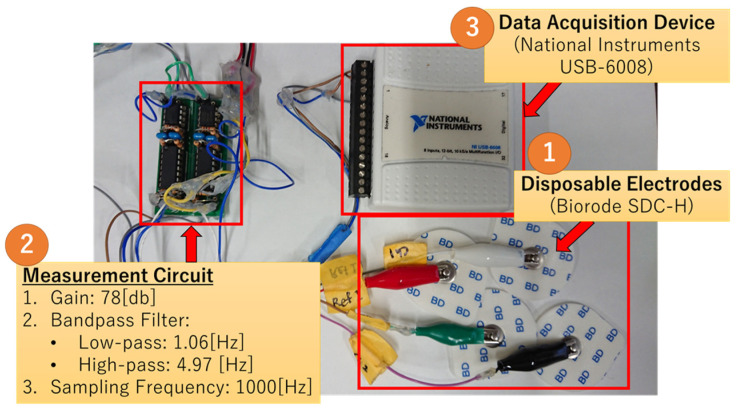
EOG measurement devices.

**Figure 3 biomimetics-08-00208-f003:**
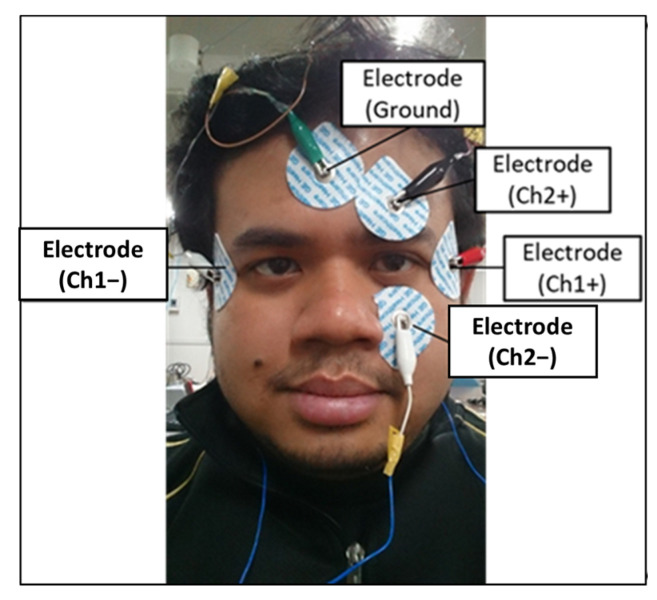
Electrode placement on the face (for vertical and horizontal eye movements).

**Figure 4 biomimetics-08-00208-f004:**
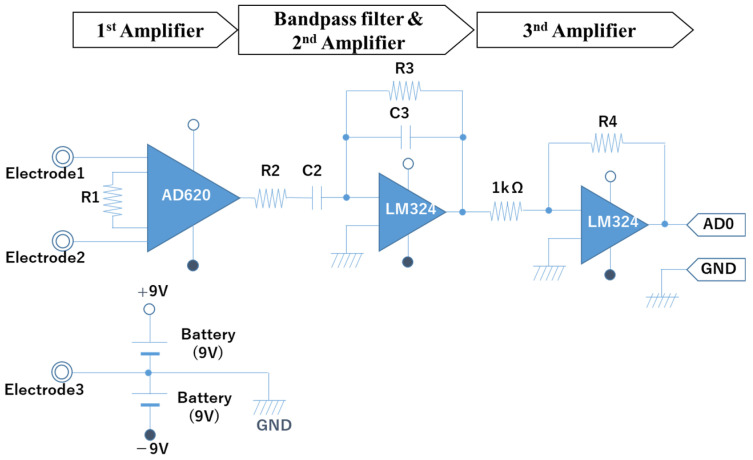
Schematic design of the bio-signal sensor.

**Figure 5 biomimetics-08-00208-f005:**
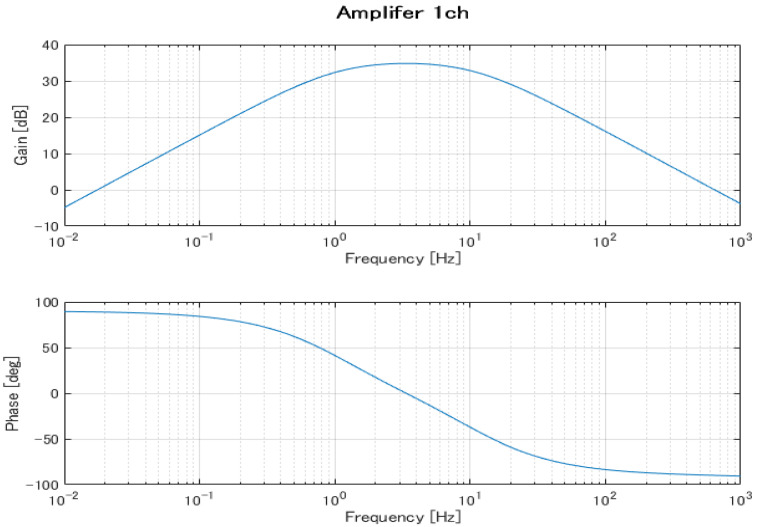
Frequency characteristics of the Ch1 amplifier circuit.

**Figure 6 biomimetics-08-00208-f006:**
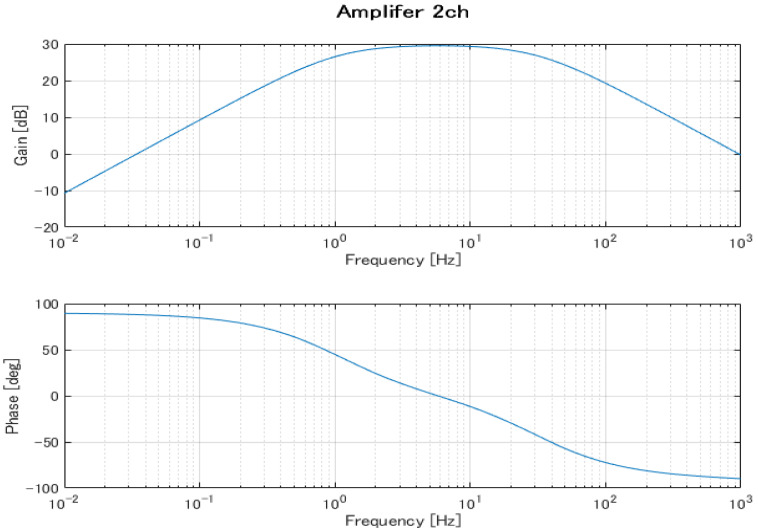
Frequency characteristics of the Ch2 amplifier circuit.

**Figure 7 biomimetics-08-00208-f007:**
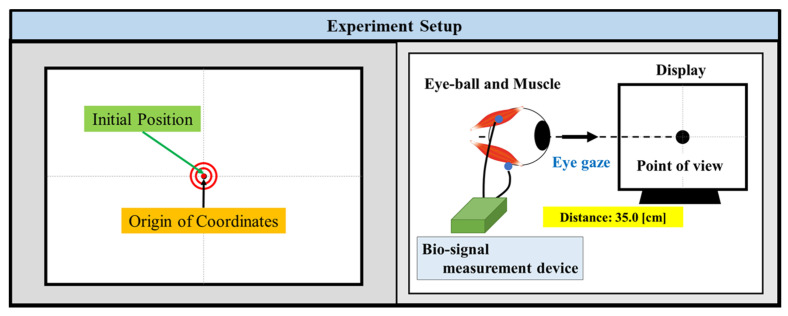
Configuration of the gaze interface and eye position.

**Figure 8 biomimetics-08-00208-f008:**
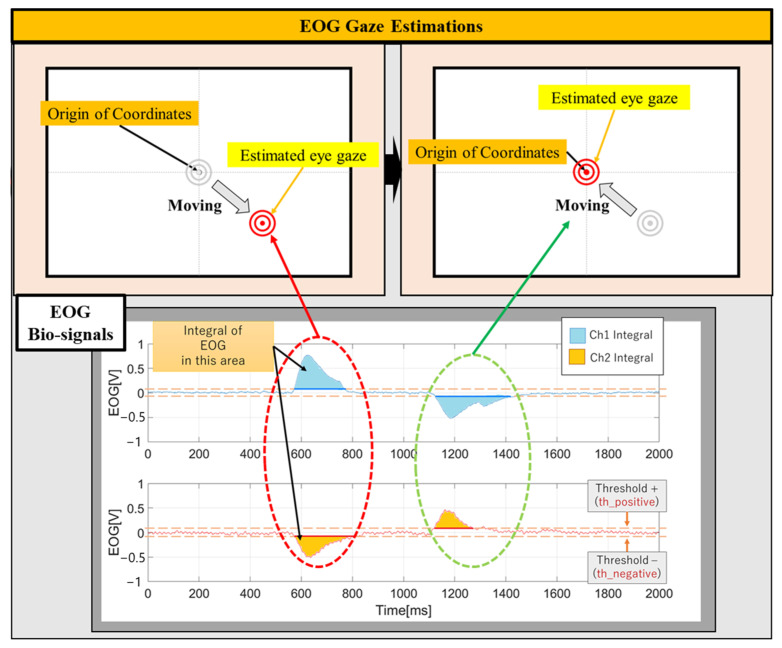
EOG gaze estimation.

**Figure 9 biomimetics-08-00208-f009:**
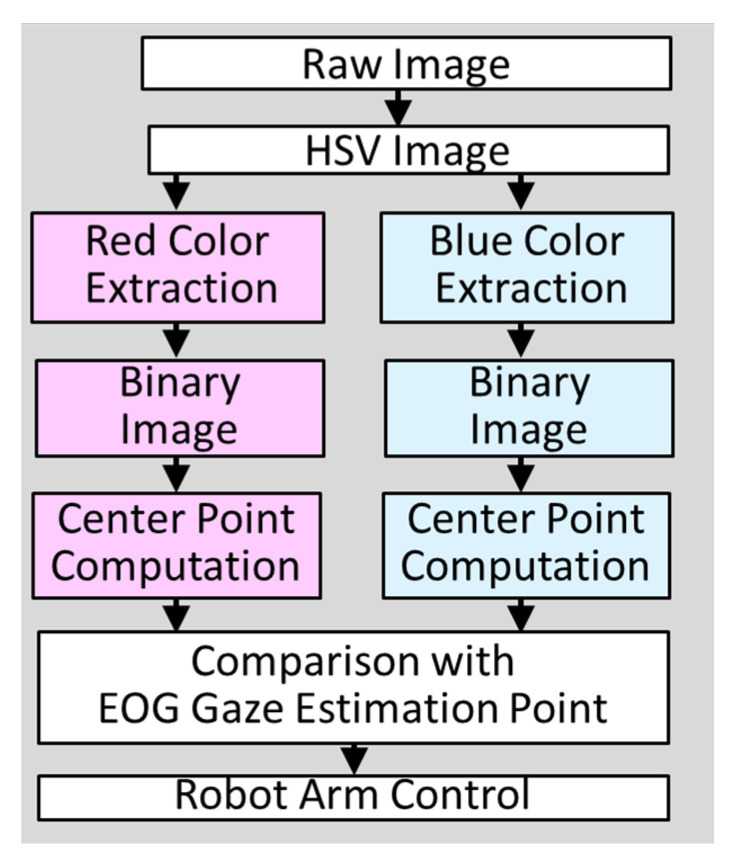
The algorithm for camera object recognition.

**Figure 10 biomimetics-08-00208-f010:**
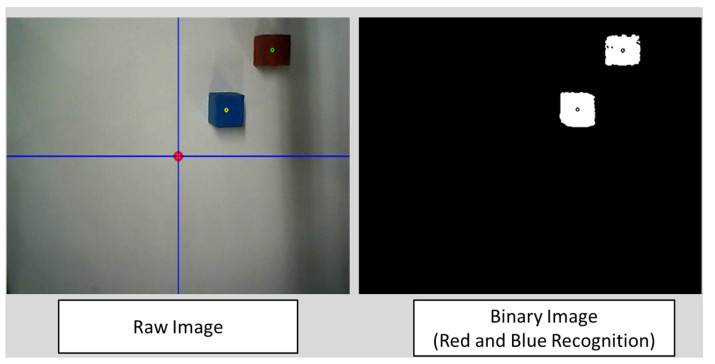
Comparison of image between raw images and binary images for object recognition.

**Figure 11 biomimetics-08-00208-f011:**
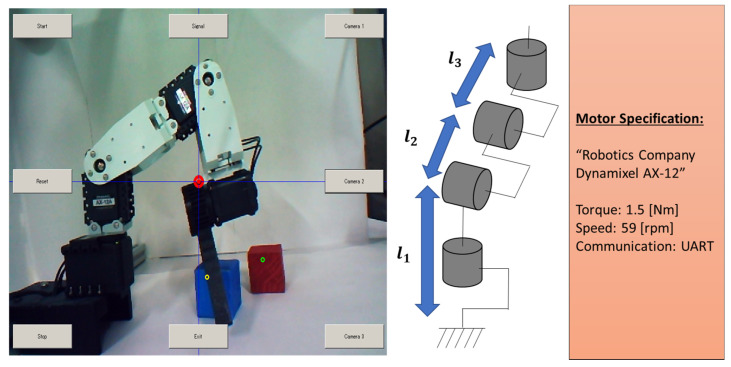
Experiment setup: 4-DoF robot arm.

**Figure 12 biomimetics-08-00208-f012:**
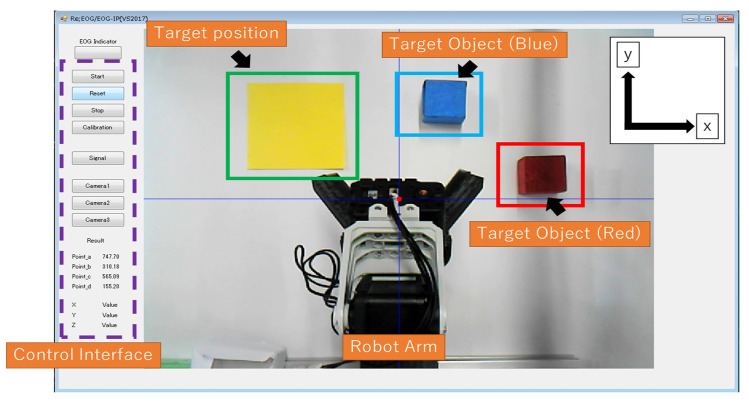
Two camera images: (**Up**) the top camera image; (**Down**) the side camera image.

**Figure 13 biomimetics-08-00208-f013:**
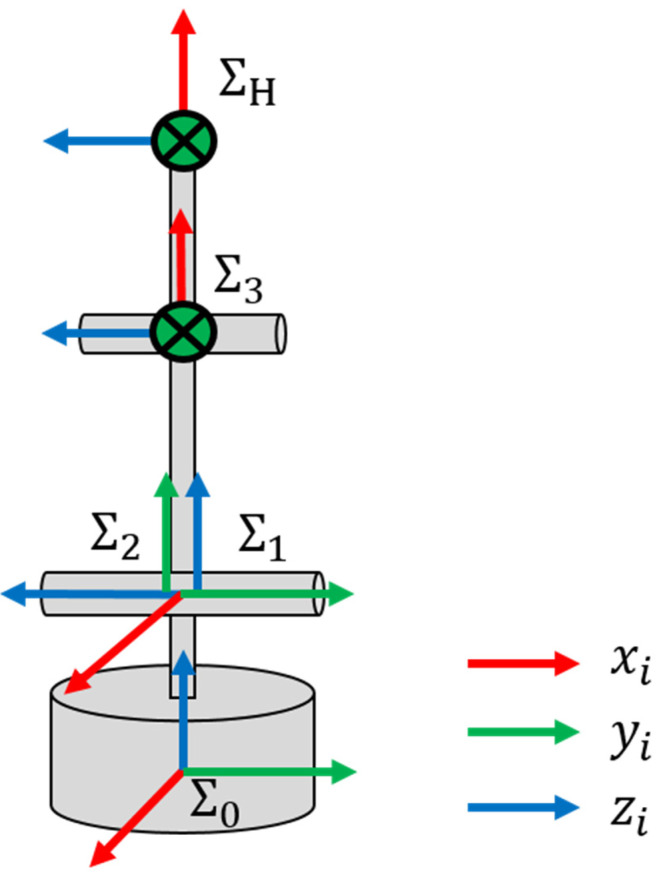
Coordinate systems of the manipulator.

**Figure 14 biomimetics-08-00208-f014:**
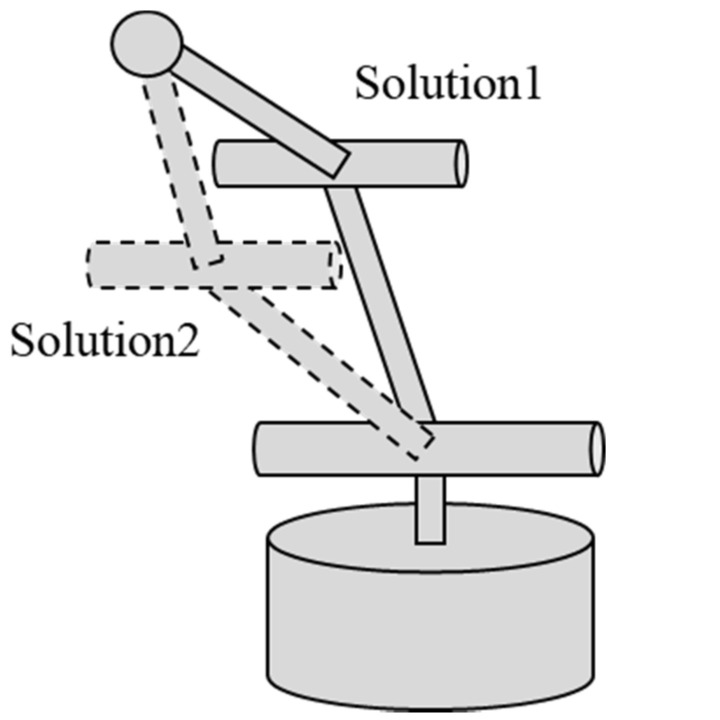
Solutions of inverse kinematics.

**Figure 15 biomimetics-08-00208-f015:**
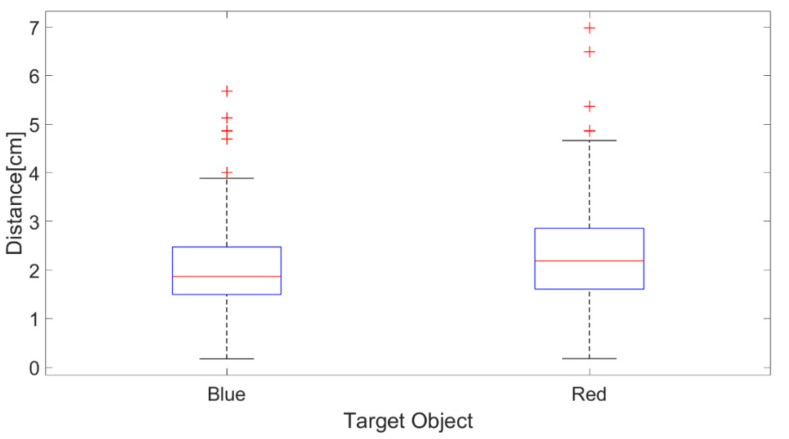
The distance errors of the EOG gaze estimations for blue and red cubes. Distance errors beyond the whiskers are displayed using red + marks.

**Figure 16 biomimetics-08-00208-f016:**
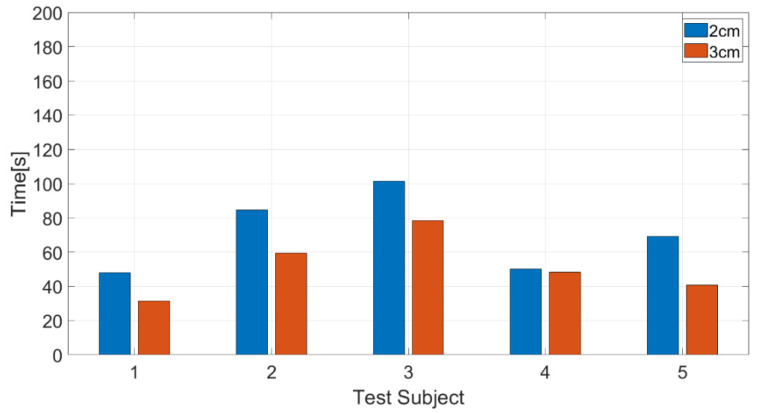
The average task completion time for five test subjects.

**Table 1 biomimetics-08-00208-t001:** The amplification values and gains for EOG and EMG measurements.

Content	1st Amplifier	Bandpass 2nd Amplifier					3rd Amplifier	Final Output	
	Capacitor Value R1 [Ω]	Filter Type	Cut-Off Frequency [Hz]	Resistor Value (R2/R3) [Ω]	Capacitor Value [F]	[uF]	Resistor Value (R5/R4) [Ω]	Amplification Factor	Gain [db]
Set Value	1000	High-pass	1.06	15,000	0.00001	10	750,000	7912	78
		Low-pass	4.97	3200	0.00001	10	1000		
Amplification factor	49.4	0.21			750.0				

**Table 2 biomimetics-08-00208-t002:** The task completion time of five subjects.

		Task Completion Time s	
Subject	Trial	Distance Threshold 2 cm	Distance Threshold 3 cm
1	1	36.25	33.34
	2	54.79	27.43
	3	52.63	33.72
2	1	150.50	79.72
	2	65.27	31.57
	3	38.04	66.83
3	1	185.31	85.80
	2	45.77	86.88
	3	73.27	62.88
4	1	49.56	53.73
	2	47.49	47.25
	3	53.47	43.65
5	1	112.03	37.93
	2	50.13	41.93
	3	45.29	42.71

## Data Availability

The data presented in this study are available on request from the corresponding author. The data are not publicly available due to privacy restrictions.

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
