# Peer review of "Object Grasp Control of a 3D Robot Arm by Combining EOG Gaze Estimation and Camera-Based Object Recognition"

_biomimetics, 2023, doi:10.3390/biomimetics8020208_

Round 1

Reviewer 1 Report

The paper presents a research of the object-grasping of 3D robot arm control based on EOG gaze estimation for people with tetraplegia. It is a topic of interest to the researchers in the related areas but the paper needs very significant improvement before acceptance for publication. My detailed comments as follows:

1. The experimental results have not been compared with other methods in depth, so the author may consider this suggestion to increase the persuasiveness.

2. Figure 13 would have been clearer.

3. The author can study deep learning methods to solve relevant problems.

Some sentences in the article are not clear, please revise carefully.

Author Response

  1. The experimental results have not been compared with other methods in depth, so the author may consider this suggestion to increase the persuasiveness.

The authors have so far worked on a method for estimating the position of an object by estimating where the operator is looking directly from the electrooculography (1)-(5). We decided to use image processing to simplify the method of estimating the coordinate position using electrooculography. To successfully grasp an object using robot arm, the robot gripper needs to move to the center point of the object. The image processing assist the gaze estimation to select the center point.

There is a difference in estimating the direction of movement from the initial position, so it seems difficult to compare the results.

  • Affine Transform to Reform Pixel Coordinates of EOG Signals for Controlling Robot Manipulators Using Gaze Motions, M. I Rusydi, M. Sasaki, S. Ito, Sensors, Vol. 14, No. 6, pp. 10107-10123, doi:10.3390/s140610107, 2014.
  • Calculate Target Position of Object in 3-Dimensional Area Based on the Perceived Locations Using EOG Signals, Muhammad Ilhamdi Rusydi1, Minoru Sasaki, Satoshi Ito, Journal of Computer and Communications, 2014, 2, pp.53-60.
  • Estimation of Viewpoint Direction and Viewing Angle Using EOG, Minoru Sasaki, Naoya Ozeki, Satoshi Ito, Harrison Ngetha, Journal of Applied Sciences, Engineering and Technology for Development, Vol. 3, No. 1, 1 – 10, 3 July 2018.
  • 24-Gaze-Point Calibration Method for Improving the Precision of AC-EOG Gaze Estimation, Muhammad Syaiful Amri bin Suhaimi, Kojiro Matsushita, Minoru Sasaki and Waweru Njeri, MDPI, Sensors, 22 August 2019.
  • Mapping 3 EMG signals generated by Human Elbow and Shoulder Movements to 2 DoF Upper-Limb Robot Control. Pringgo Widyo Laksono1, Kojiro Matsushita1, Muhammad Syaiful Amri bin Suhaimi, Takahide Kitamura, Waweru Njeri, Joseph Muguro and Minoru Sasaki1, Robotics, MDPI, 2020.

  1. Figure 13 would have been clearer.

Figure 13 shows the inverse kinematics of the robot. According to the reviewer’s comment we add the explanation of the inverse kinematics of the robot.

  1. The author can study deep learning methods to solve relevant problems.

In general, deep learning is one approach to this problem, but it can obscure the nature of dataset problems and challenges in image processing (6). By performing basic image processing with the first approach, the authors decided to examine what kind of machine learning application would be appropriate next by clarifying the next problem to be improved.

(6) Construction of an Environmental Map including Road Surface Classification Based on a Coaxial Two-Wheeled Robot Minoru Sasaki, Eita Kunii, Tatsuya Uda, Kojiro Matsushita, Joseph K. Muguro, Muhammad Syaiful Amri bin Suhaimi and Waweru Njeri, Journal of Sustainable Research in Engineering Vol. 5 (3) 2020, 159 – 169.

Reviewer 2 Report

This paper proposes a method for achieving accurate object grasping with a 3D robot arm controlled by Electrooculography (EOG) signals. The system combines EMG gaze estimation and object recognition of camera image processing to compensate for lost information and increase spatial accuracy. The proposed system enables highly accurate object grasping with a stable and fast selection process.

1. To improve this paper, the sentence structure needs to be rearranged, and table 1 needs to be repositioned with added internal borders.

2. Additionally, there needs to be more detailed explanations of figures 5, 6, 7, and 8 in the sentence, and the captions for figures 7 and 8 need to be modified for clarity.

3. The author is recommended to provide a detailed and comprehensive explanation of the reliability and effectiveness of the entire method and make necessary modifications. Moreover, it is essential to discuss the pros and cons of the system.

4. The paper explains how to estimate distance using EOG with two channels and how to set up the threshold value. However, there are no demonstration videos provided. To improve the figure's layout, it is suggested to make one row in figure 11 instead of two and explain the figures before presenting them. The position of the figures should be modified accordingly. Figure 13 could be placed in the supplementary section or redrawn without an equation.

5. What is the cause of the distance error that occurs in EOG Gaze Estimation?

Please checking spaces between words.

Author Response

  1. To improve this paper, the sentence structure needs to be rearranged, and table 1 needs to be repositioned with added internal borders.

According to the reviewer’s comment we have revised the sentence structure.

  1. Additionally, there needs to be more detailed explanations of figures 5, 6, 7, and 8 in the sentence, and the captions for figures 7 and 8 need to be modified for clarity.

We have revised the description of the paper according to the referee's instructions.

  1. The author is recommended to provide a detailed and comprehensive explanation of the reliability and effectiveness of the entire method and make necessary modifications. Moreover, it is essential to discuss the pros and cons of the system.

The paper’s methodology is based on the previous work done by the author. The authors have worked on a method for gaze estimation using EOG to target a location (1)-(4). Then, further explanation of the inverse kinematic of the robot has been included.

  • Affine Transform to Reform Pixel Coordinates of EOG Signals for Controlling Robot Manipulators Using Gaze Motions, M. I Rusydi, M. Sasaki, S. Ito, Sensors, Vol. 14, No. 6, pp. 10107-10123, doi:10.3390/s140610107, 2014.
  • Calculate Target Position of Object in 3-Dimensional Area Based on the Perceived Locations Using EOG Signals, Muhammad Ilhamdi Rusydi1, Minoru Sasaki, Satoshi Ito, Journal of Computer and Communications, 2014, 2, pp.53-60.
  • Estimation of Viewpoint Direction and Viewing Angle Using EOG, Minoru Sasaki, Naoya Ozeki, Satoshi Ito, Harrison Ngetha, Journal of Applied Sciences, Engineering and Technology for Development, Vol. 3, No. 1, 1 – 10, 3 July 2018.
  • 24-Gaze-Point Calibration Method for Improving the Precision of AC-EOG Gaze Estimation, Muhammad Syaiful Amri bin Suhaimi, Kojiro Matsushita, Minoru Sasaki and Waweru Njeri, MDPI, Sensors, 22 August 2019.

We also have included the pro and cons of the system.

The pro:

  1. The system enables remote control of a robot arm using the EOG method.
  2. The implementation of the simple image processing method improves the EOG gaze estimation targeting the center point of the object.
  3. The robot gripper able to grab the target object successfully for displacement task.

The cons:

  1. The user is constrained to stay still and positioned 0.35 [m] from the computer display.
  2. The condition of the captured image from camera can affect the accuracy of the image processing. As examples, the image brightness from room lighting and image pixel quality.

  1. The paper explains how to estimate distance using EOG with two channels and how to set up the threshold value. However, there are no demonstration videos provided. To improve the figure's layout, it is suggested to make one row in figure 11 instead of two and explain the figures before presenting them. The position of the figures should be modified accordingly. Figure 13 could be placed in the supplementary section or redrawn without an equation.

It has been revised according to the referee's instruction. Figure 13 has been rewritten.

  1. What is the cause of the distance error that occurs in EOG Gaze Estimation?

The electrooculogram changes with the movement of the eye, and the impedance changes depending on the dryness between the electrodes and the skin. There are individual differences in skin conditions. In addition, the electrooculography zero point and the initial position of the eyeball change subtly. For that reason, a certain threshold is set to ensure stable operation, but this may also cause errors.

Round 2

Reviewer 2 Report

The reviewer is curious if there are any similar works to the paper, particularly the 3D gaze estimation for grasping paper by Li et al. (2017) titled "3-D-gaze-based robotic grasping through mimicking human visuomotor function for people with motion impairments" published in IEEE Transactions on Biomedical Engineering.

The reviewer would like to know if there are any state-of-the-art works that compare to the authors' work and, if so, how their work differs.

If there are similar systems, the authors can compare the reliability and effectiveness of their method or system with those of the state-of-the-art.

The reviewer acknowledges the authors' contribution to the new architecture of the system but notes that inverse kinematics (traditional) is not their contribution.

Finally, the reviewer suggests that the authors check the font in Figure 16 and the arrangement in Table 2 to ensure they comply with the guidelines.

It is well-written.

Author Response

Responses to reviewers

The reviewer is curious if there are any similar works to the paper, particularly the 3D gaze estimation for grasping paper by Li et al. (2017) titled "3-D-gaze-based robotic grasping through mimicking human visuomotor function for people with motion impairments" published in IEEE Transactions on Biomedical Engineering.

The reviewer would like to know if there are any state-of-the-art works that compare to the authors' work and, if so, how their work differs.

If there are similar systems, the authors can compare the reliability and effectiveness of their method or system with those of the state-of-the-art.

The reviewer acknowledges the authors' contribution to the new architecture of the system but notes that inverse kinematics (traditional) is not their contribution.

Regarding the conventional research raised by the reviewer, "3-D-Gaze-Based Robotic Grasping Through Mimicking Human Visuomotor Function for People with Motion Impairments", the author has thoroughly read and examined the paper. We are challenging robot control in a 3D environment using eye gaze estimation, but our approach is fundamentally different.

The current authors proposed a method for estimating the position of an object by estimating where the operator is looking directly from the Electrooculography (EOG). Our proposed technique is to acquire eye movement by using bio-signal sensors to get the EOG eye’s signal. The signal is then analysed to get the gaze estimation in 2D coordinates. By implementing two cameras in a robotic system, we can achieve 3D control for the robotic arms using EOG-based gaze estimation. As the EOG-based gaze estimation have an accuracy issue (1) – (4) and robot grasping requires precise control, we proposed the image processing method to assist the EOG gaze estimation accuracy.

  • Affine Transform to Reform Pixel Coordinates of EOG Signals for Controlling Robot Manipulators Using Gaze Motions, M. I Rusydi, M. Sasaki, S. Ito, Sensors, Vol. 14, No. 6, pp. 10107-10123, doi:10.3390/s140610107, 2014.
  • Calculate Target Position of Object in 3-Dimensional Area Based on the Perceived Locations Using EOG Signals, Muhammad Ilhamdi Rusydi1, Minoru Sasaki, Satoshi Ito, Journal of Computer and Communications, 2014, 2, pp.53-60.
  • Estimation of Viewpoint Direction and Viewing Angle Using EOG, Minoru Sasaki, Naoya Ozeki, Satoshi Ito, Harrison Ngetha, Journal of Applied Sciences, Engineering and Technology for Development, Vol. 3, No. 1, 1 – 10, 3 July 2018.
  • 24-Gaze-Point Calibration Method for Improving the Precision of AC-EOG Gaze Estimation, Muhammad Syaiful Amri bin Suhaimi, Kojiro Matsushita, Minoru Sasaki and Waweru Njeri, MDPI, Sensors, 22 August 2019.

As for the conventional research, the author is using camera-based gaze-tracking. A glasses-type IR camera is used to obtain the left and right eye’s pupil images. By analysing and vectoring the left and right pupils’ location from the images (gaze vector method), the author can determine the eye’s depth perception. Thus, able to estimate the eye gaze in a 3D environment. The eye gaze estimation based on depth perception is then used to control a robot arm in a 3D environment.

Finally, the reviewer suggests that the authors check the font in Figure 16 and the arrangement in Table 2 to ensure they comply with the guidelines.

It has been revised according to the referee's instructions. Figure 16 has been checked for the font and Table 2 has been checked for arrangement. Both Figure 16 and Table 2 format also rechecked.
